# Advances in Ferritin Physiology and Possible Implications in Bacterial Infection

**DOI:** 10.3390/ijms24054659

**Published:** 2023-02-28

**Authors:** Clemens M. Gehrer, Anna-Maria Mitterstiller, Philipp Grubwieser, Esther G. Meyron-Holtz, Günter Weiss, Manfred Nairz

**Affiliations:** 1Department of Internal Medicine II, Infectious Diseases, Immunology, Rheumatology, Medical University of Innsbruck, 6020 Innsbruck, Austria; 2Laboratory of Molecular Nutrition, Faculty of Biotechnology and Food Engineering, Technion-Israel Institute of Technology, Haifa 32000, Israel; 3Christian Doppler Laboratory for Iron Metabolism and Anemia Research, Medical University of Innsbruck, 6020 Innsbruck, Austria

**Keywords:** ferritin, compartmentalization, immune metabolism, exosome, nutritional immunity, regulation, *Salmonella* Typhimurium, iron, macrophage, liquid-liquid phase separation

## Abstract

Due to its advantageous redox properties, iron plays an important role in the metabolism of nearly all life. However, these properties are not only a boon but also the bane of such life forms. Since labile iron results in the generation of reactive oxygen species by Fenton chemistry, iron is stored in a relatively safe form inside of ferritin. Despite the fact that the iron storage protein ferritin has been extensively researched, many of its physiological functions are hitherto unresolved. However, research regarding ferritin’s functions is gaining momentum. For example, recent major discoveries on its secretion and distribution mechanisms have been made as well as the paradigm-changing finding of intracellular compartmentalization of ferritin via interaction with nuclear receptor coactivator 4 (NCOA4). In this review, we discuss established knowledge as well as these new findings and the implications they may have for host–pathogen interaction during bacterial infection.

## 1. Introduction

Bacterial infection has been a major public health concern for a long time. Accordingly, a recent study has estimated that deaths associated with 33 common bacterial genera or species would rank as the second leading cause of death globally in 2019 [1,2]. During infection, pathogens and the host are entangled in a fight for their respective lives, whereby the host as well as the pathogen use intricate mechanisms to ensure their survival, such as the complement system, quorum sensing, which is the ability of cells, e.g., bacteria, to adapt their metabolism in response to cell population density [3,4,5]. Furthermore, the host and pathogens both compete for nutrients during infection. Thereby, the host attempts to deprive invading microorganisms from essential nutrients by a distinct set of mechanisms, while the pathogen tries to circumvent these efforts. This part of the non-adaptive immune response, termed nutritional immunity, has been shown to be essential in fighting off pathogens [6,7,8,9]. The research concerning this matter has gained momentum in recent years. In this context, iron is regarded as one of the central nutrients involved in nutritional immunity, because it is essential for nearly all forms of life [10]. The constant demand for iron is based on its pivotal role in a wide array of enzymatic processes. For example, DNA synthesis, reactive oxygen species (ROS) defense and energy metabolism require iron due to its advantageous redox properties. However, iron is not only the boon, but also the bane of cellular metabolism, because it has the ability to produce ROS via Fenton chemistry. Therefore, iron is stored intracellularly in the protein ferritin, which can bind large quantities of iron and thus diminish the generation of ROS.

Recent insights demonstrated that intracellular ferritin is present in its own membrane-less compartment inside the cytoplasm separated via liquid-liquid phase separation. Furthermore, increasing evidence of the mechanisms of ferritin trafficking has accumulated in the last decade. In this review, we discuss the currently known mechanisms of mammalian ferritin’s function, regulation, compartmentalization and trafficking, as well as the possible implications they might have in host–pathogen interaction. Thereby, we put the focus on infection with the intracellular bacterium *Salmonella enterica* serovar Typhimurium (*S*.Tm), but we will also discuss other intracellularly occurring pathogens. We will not discuss the function of prokaryotic mini- and maxi-ferritins, because a comprehensive overview has recently been provided by Bradley et al. [11].

### 1.1. Mammalian Ferritin’s Morphology and Function

Mammalian ferritin is a spheric protein, which can consist of two types of subunits, namely the ferritin H-chain (FTH, ~21 kDa in humans) and the ferritin L-chain (FTL, ~19 kDa in humans) [12]. These two subunits assemble into hollow 24-meric nanocages in variable ratios. Therefore, the molecular weight of the assembled protein is variable depending on the tissue specific H/L-ratio, but is generally around 480 kDa for the iron-free protein, named apoferritin. Although the ferritin subunits are able to assemble to homopolymers, most ferritin is found as a heteropolymeric protein in vivo. With this in mind, ferritin isolated from the brain, heart, kidney, pancreas, muscle, thymus and red blood cells is rich in FTH, while ferritin from liver and spleen is rich in FTL [12,13,14,15,16]. A special case is serum ferritin, which has been proposed to contain only minute amounts of FTH, and which is thought to contain only little amounts of iron, with one study in humans even proposing that it might represent apoferritin [17,18,19,20,21]. The reasons for the tissue-specific subunit composition are hitherto unclear. However, it is generally thought that H-rich ferritin occurs in tissues with a higher demand for anti-oxidative capacity and L-rich ferritin occurs in tissues, which are more involved in the storage of iron [22,23,24]. This idea arises from the distinct functions of these two proteins. FTH possesses a ferroxidase active site, which catalyzes the oxidation of ferrous to ferric iron (vide infra) [15]. In contrast, FTL does not have this catalytic site, but has negatively charged amino acid residues, which serve as nucleation sites for the formation of the iron core within the ferritin complex [25,26]. Furthermore, FTL supports the ferroxidase activity of the FTH by increasing the iron turnover at the catalytic center [27].

Once assembled, mammalian ferritin forms two kinds of channels, localized between the subunits connecting the environment with the cavity of ferritin. Thereby, there are eight channels with a three-fold symmetry and six channels with a four-fold symmetry. The three-fold channels are hydrophilic, as they are lined with six negatively charged residues and are regarded as the entry point of iron into ferritin [28,29,30,31]. In contrast, the four-fold channels are hydrophobic and not thought to transport iron, but are still involved in the incorporation of iron into ferritin. This is because changes in the residues of the four-fold channel interfere with ferritin’s ability to deposit iron inside its core [32,33,34]. The suggested function of the four-fold channels is to transfer protons, generated during the formation of the core, from the inside ferritin to the environment [29,35]. However, this proposed function comes from in-silico analyses of the electrostatic properties of ferritin, but experimental data confirming this function are lacking. Furthermore, the amino acids adjacent to the four-fold channel are important for the assembly and the stability of ferritin [33,34]. In addition to the inter-subunit channels, there is also a small channel in each FTH, which connects the outside of ferritin with the ferroxidase site [36]. This channel was proposed to be important for oxygen transport to the ferroxidase site, but this could not be confirmed by experimental data, as a change in the key amino acid Tyr 29 did not result in an altered iron incorporation into ferritin [37,38]. A detailed depiction of ferritin structure is provided by Ebrahimi et al. [39].

The main function of ferritin is regarded to be the intracellular detoxification of iron through the incorporation of the metal and the concomitant prevention of ROS formation [40]. Iron is taken up via the three-fold pore and then transported to the ferroxidase active site [41]. There, it is oxidized under consumption of oxygen or hydrogen peroxide before it starts forming the core by binding to specific nucleation sites on the inside of the ferritin shell [25,26]. The rates of oxidation with H_2_O_2_ are much faster than with oxygen, ~3-times faster in H-rich ferritin and ~120-times faster in L-rich ferritin [27]. Furthermore, the mineral core can facilitate the oxidation and subsequent core formation enzymatically, when it reaches a size of approximately 200 iron atoms [27,42,43]. These reactions can be summarized in three equations [27]:2Fe^2+^ + O_2_ + 4H_2_O → 2Fe(O)OH(core) + H_2_O_2_ + 4H^+^ (ferroxidation reaction)(1)
4Fe^2+^ + O_2_ + 6H_2_O → 4Fe(O)OH(core) + 8H^+^ (mineral surface reaction)(2)
2Fe^2+^ + H_2_O_2_ + 2H_2_O → 2Fe(O)OH(core) + 4H^+^ (detoxification reaction)(3)

As the sum of reactions (1) and (3) is the same as reaction (2), these reactions cannot be distinguished [27]. The end product of these reactions is a core, which has been shown to mainly consist of ferrihydrite and hematite with traces of magnetite and maghemite [44]. In addition to the ability of ferritin to detoxify H_2_O_2_, the core of ferritin has been proposed to harbor superoxide dismutase activity [45].

Aside from its function in ROS detoxification and iron storage, ferritin is involved in immunomodulation. In this context, FTH was found to act as a pro-inflammatory signal in liver cells independent of its iron content via activation of the nuclear factor (NF)-ĸB-pathway [46]. However, studies in other cells could show that binding of FTH correlates with impaired B-cell maturation and immunoglobulin production as well as reduced T-cell proliferation [47,48,49]. Furthermore, FTH has pro- as well as anti-inflammatory effects depending on the cell type. CD8+ T-cells express interferon-γ (IFN-γ) upon stimulation with FTH. In contrast, regulatory T-cells, in contact with dendritic cells, prestimulated with FTH, secrete the anti-inflammatory cytokine interleukin (IL)-10 [50]. However, the role of ferritin in immunomodulation is also under debate as some findings have been controversial [51,52]. One study has shown that FTH is necessary for disease tolerance in a cecal ligation and puncture sepsis model, but did not affect cytokine expression [52], while another study using the same sepsis model showed an *FtH* knockout to enhance survival in mice via a blunted immune response [51]. However, while one study showed that the addition of exogenous horse spleen apoferritin reduced septic lethality [52], the other study showed apoferritin or recombinant human FTL mitigated LPS-mediated macrophage activation [51], which indicates an immunosuppressive function of ferritin. In contrast, a recent study could show that administration of horse spleen holoferritin results in hepatic as well as systemic inflammation, which is mediated via the ferritin receptor Scavenger receptor class A member (SCARA)1 and subsequent neutrophil extracellular trap formation [53]. These contrasting findings might be explained by the iron content of ferritin or changes in the protein during the removal of iron from ferritin by the manufacturer and by the different models used. Of note, another study found that macrophage specific depletion of FTH resulted in impaired control of Salmonella infection in iron-loaded mice which could be traced back to induction of inflammasome activation and IL-1β formation. Accordingly, Caspase-1 inhibition or anti-IL-1 treatment could overcome this survival disadvantage indicating that ferritin is an important radical detoxifying molecule in the setting of pathologic inflammation and infection [54]. Furthermore, this study also found that peritoneal macrophages lacking FTH demonstrate a higher baseline secretion of pro-inflammatory cytokines, which is independent of iron stimulation [54].

Nevertheless, these studies indicate there to be an immunomodulatory function of ferritin, albeit still unresolved whether it acts pro- or anti-inflammatory or even both.

### 1.2. Regulation of Ferritin

The regulation of ferritin expression is very complex and involves transcriptional and post-transcriptional mechanisms [10]. An extensive review of factors influencing ferritin expression has been given by Torti et al. [55]. In that review, they describe a wide array of transcriptional and post-transcriptional regulation through iron, oxidative stress, inflammatory signals, hormones, growth factors, second messengers and cell differentiation. Herein, we will focus on the former three, as they seem the most important ones in the context of our review. A simplistic schematic depiction of the mechanisms explained below is provided in Figure 1.

Generally speaking, the regulation of ferritin is regarded to occur mostly on a post-transcriptional level by facilitating the transfer of ferritin mRNA from a mostly inactive pool to polyribosomes in order to initiate the translation of the protein [56]. Thereby, ferritin translation is regulated by iron through iron regulatory protein (IRP) 1 and 2 [55,57]. When the cellular iron content is low, the IRPs bind to the iron responsive element (IRE) in the 5′ untranslated region of the H- and L-ferritin mRNA [58,59,60,61], thus preventing translation [62]. The inhibition of translation by IRP1 and presumably also IRP2 happens at the stage of initiation, as IRP1 has been shown to prevent the interaction of the eukaryotic translation initiation factor (eIF) complex eIF4F with ribosomes [63]. When there is a sufficient amount of cellular iron, IRP1 contains an iron-sulfur cluster, becoming cytoplasmic aconitase [64], and IRP2 is degraded. The latter pathway is facilitated by F-box and leucine-rich repeats protein 5 (FBXL5) in an iron and oxygen-dependent manner [65]. Consequently, in the presence of surplus iron, the IRPs move away from the IRE of ferritin mRNA, permitting the translation of the protein for iron storage. Interestingly, a recent study has shown that FTL, but not FTH, expression is post-transcriptionally repressed by eIF3, which is distinct from the regulation via the IRP system [66]. However, the binding of IRPs to IRE can be modulated via ROS and reactive nitrogen species (RNS), but the findings are controversial and subject to debate, as has been reviewed by Cairo et al. [67]. H_2_O_2_ promotes IRP1 binding to IRE, most likely through phosphorylation or an energy-dependent mechanism rather than by direct attack [68,69]. Furthermore, IRP1 activity is regarded to be increased by NO [70,71,72,73,74,75], while its effect on the IRE-binding activity of IRP2 is less clear. Some studies found IRP2 activity to be enhanced by NO [71,76], while others found a reduction in IRP2 activity [73,74,77]. In this context, Kim et al. have proposed that the distinct effect of the NO congeners NO^+^ and NO^.^ may be responsible for these paradoxical findings [74]. The distinct effects of these two nitrogen species have also been shown experimentally [78]. Furthermore, IRP1 seems to be only transiently activated by ROS and RNS with a subsequent inactivation and concomitant enhanced ferritin translation [73,79]. Besides the post-transcriptional regulation by ROS, ferritin is induced on a transcriptional level via the binding of nuclear factor erythroid 2-related factor 2 (Nrf-2) to an antioxidant responsive element (ARE) in the promoter regions of ferritin [80,81]. It is of note that most the studies on the influence of ROS/RNS on IRP1 and 2 have mostly been conducted in vitro under standard cell culture conditions, and therefore, might be of limited relevance for the physiological environment, which has roughly 3–7% oxygen [82].

Ferritin regulation has also been shown to be influenced by inflammatory stimuli other than ROS and RNS [83,84,85,86,87]. Specifically, the cytokines tumor necrosis factor α (TNF-α), IL-1β, IFN-γ, IL-6 and the anti-inflammatory IL-10 [88] induce ferritin expression. In the case of TNF-α and IFN-γ this happens on a transcriptional level, whereby H-ferritin mRNA expression is upregulated [87,89]. On the other hand, IL-1β does not increase ferritin mRNA, but facilitates ferritin translation by binding to a G and C rich region in the 5′ UTR, which is also present in the mRNAs of other acute phase proteins [84,85,87]. IL-10 also promotes ferritin expression on a post-transcriptional level, but the exact mechanism is not clear [88]. In the case of IL-6, an enhanced ferritin protein expression has been reported, but the effect on mRNA transcription was not assessed [86]. In addition to enhancing ferritin expression, TNF-α as well as IFN-γ enhance the flux of newly acquired iron into ferritin, while the overall iron uptake is reduced [87,90]. However, another study has demonstrated that the overall ferritin-bound iron is reduced upon stimulation with IFN-γ and that infection with *S.*Tm further decreases the amount of ferritin-bound iron [91,92].

Taken together, multiple mechanisms control ferritin expression and many of these are deeply involved in inflammation and infection. While post-transcriptional regulation via the IRP/IRE system is at the regulatory center of ferritin protein expression, transcriptional and IRP-independent post-transcriptional mechanisms play an important role for cellular ferritin concentration. Furthermore, the three described regulatory mechanisms influence each other, which makes it extremely difficult to attribute the observed results of previous studies to solely one stimulus.

### 1.3. Cellular Location, Trafficking and Utilization of Ferritin

Although mammalian ferritin, with the exception of mitochondrial ferritin, has been considered a cytosolic protein, a recent study found it in a separate membrane-less compartment inside the cytosol [93]. In that study, the authors defined a new role of the canonical ferritinophagy receptor nuclear receptor coactivator 4 (NCOA4). The interaction of NCOA4 and H-ferritin results in liquid-like ferritin-NCOA4 condensates, which are separated from the cytosol due to liquid–liquid phase separation. However, the finding of membrane-less ferritin aggregates has been described much earlier [94,95,96].

This condensate formation is necessary for subsequent utilization of ferritin and vesicular trafficking of the protein via the binding of Tax1-binding protein 1 (TAX1BP1) to NCOA4 [93,97]. Thereby, TAX1BP1 targets ferritin to lysosomes via classical Microtubule-associated protein 1A/1B-light chain 3 (LC3)-dependent [93] and alternative pathways, the latter being independent of Autophagy-related protein (ATG) 8 but dependent on FAK family kinase-interacting protein of 200 kDa (FIP200), TANK-binding kinase 1 (TBK1) and Phosphatidylinositol 3-kinase VPS34 (VPS34) [97]. Furthermore, the alternative pathway seems not to be restricted to NCOA4 as a substrate. Rather, the proteins involved are also important for the turnover of another TAX1BP1 substrate, namely Next to BRCA1 gene 1 protein (NBR1) [98].

This mechanism of lysosomal ferritin degradation is also regulated by iron via NCOA4 [99]. Under iron-rich conditions, NCOA4 binds iron and interacts with the E3 protein ligase HECT domain and RCC1-like domain-containing protein 2 (HERC2), which leads to the subsequent degradation of NCOA4 via the proteasome [99]. Consequently, ferritin shuttling to the lysosome decreases.

The trafficking of ferritin to the lysosome and its subsequent degradation is termed ferritinophagy. This process appears to be the most important way to release ferritin-bound iron, which has been shown during infection [99,100,101]. However, it is speculated, that other mechanisms, e.g., reductive iron release, may play a role in the cellular environment as well [102,103]. Hitherto, the proposed mechanisms have only been shown in cell-free systems due to difficulties in creating appropriate experimental setups in a living system. Nevertheless, ferritin has also been shown to be degraded via the proteasome and that iron is released prior to degradation [104,105]. In conclusion, there may be several mechanisms for utilization of ferritin-bound iron. However, ferritinophagy is not only the best characterized but presumably also the most important one.

In the lysosome, the ferritin core dissolves due to the acidic environment (pH ~ 4.5–5.0) [106], while the protein is degraded by lysosomal proteases [107]. Thereby, the acidification is necessary for iron release of ferritin, while the proteolytic degradation is not [107]. After the dissolving/dissolution of the ferritin core, lysosomal ferrireductases six-transmembrane epithelial antigen of prostate 3 (STEAP3) and Cyb561a3 (AKA lysosomal cytochrome B, LcytB) reduce Fe^3+^ to Fe^2+^, which is then transferred to the cytosol via Natural resistance-associated macrophage protein 1 (NRAMP1), Divalent metal transporter 1 (DMT1), also known as SLC11A2 or NRAMP2, and Transient receptor potential channel mucolipin 1 (TRPML1) [108,109,110,111,112].

However, the trafficking of ferritin to the lysosome is not only important for cellular iron utilization, but also for the secretion of ferritin [21,113]. In the lysosome, ferritin is processed resulting in truncated FTL-subunits, which are called serum-subunits (S-subunit), as they are predominantly found in serum ferritin [21]. Hence, the presence of the S-subunit in serum indicates that ferritin is subsequently released via non-classical lysosomal secretion [21]. It has also been proposed that FTL, found in serum, is secreted via the classical Golgi-dependent pathway [114]. However, this finding could not be reproduced by another study [113]. Furthermore, cells secrete ferritin via the multivesicular body (MVB)–exosome pathway, whereby this process is likely being regulated via IRE/IRP-dependent translation of CD63 [113,115,116]. In addition to that, ferritin has also been shown to exit the cell via secretory autophagy upon endomembranous damage in a Tripartite motif-containing protein 16 (TRIM16), Vesicle-trafficking protein SEC22b (Sec22b) and galectin-8 dependent manner, but seemingly independent of NCOA4 [117]. Taken together, several mechanisms exist for the secretion of ferritin. However, it remains unknown whether these mechanisms are relevant in infections and for nutritional immunity.

In humans, it has been shown that cells can take up extracellular ferritin directly via T-cell immunoglobulin and mucin domain (TIM) 1, transferrin receptor 1 (TFR1) and SCARA5 [118,119,120,121]. A recent study demonstrated that other members of the SCARA family, namely SCARA1 and SCARA2, are also able to bind human ferritin, albeit to a lesser degree than SCARA5, with SCARA2 only showing very weak binding [121]. Furthermore, it has also been shown in mice that cells can acquire ferritin via TIM2 and SCARA5, which has first been identified as a ferritin receptor in mice [122,123]. Thereby, TIM1, TIM2 and TFR1 act as FTH receptors [118,119,122], while SCARA5 supposedly binds FTL [120,123]. However, a recent study proposed that SCARA1, 2 and 5 are able to bind both FTL and FTH [121]. Additionally, ferritin can enter cells by using extracellular vesicles as a vehicle [116]. Hitherto, the mechanism behind the uptake of ferritin-containing exosomes has not been investigated, but might be facilitated by receptors, e.g., TIM-4 recognizes phosphatidylserines, which are usually enriched in exosomes [124,125,126]. This binding may result in internalization of the exosome or the fusion with the plasma membrane [127]. A graphic depiction of the described mechanisms is provided in Figure 2.

In conclusion, numerous pathways for the uptake, intracellular trafficking and release of ferritin have been described. Therefore, it may be feasible to acknowledge ferritin as a cellular iron import and export protein. However, the biological relevance of ferritin as a systemic iron transport protein is under discussion. For example, under physiologic conditions, only a small amount of ferritin is found in serum and this ferritin is less iron loaded in comparison with intracellular ferritin [21]. Nevertheless, the saturation of serum ferritin with iron in healthy human subjects has been estimated by different investigators to be roughly between 24% [128] and 50% [129]. Furthermore, we calculated the ferritin saturation of human subjects with a normal iron status from a third study [130] by using a hypothetical molecular weight of 450 kDa and a maximum loading capacity of 4500 iron atoms per ferritin, which would apply to 100% saturation. This resulted in roughly 27% saturation. In other words, human serum ferritin has been found to contain on average roughly between 1000–2000 iron atoms per ferritin, which contrasts the general sentiment of serum ferritin containing only small amounts of iron. In addition to that, a study investigating the development of *Tfr1* knockout mice embryos suggests that ferritin functions as a cell type-specific iron transport protein during organogenesis [123]. This finding is in accordance with an essential role of FTH in embryonic development [131]. However, ferritin’s function in systemic iron metabolism in adult mammals may be less important in homeostatic conditions [132].

### 1.4. General Aspects of Nutritional Immunity

The biological relevance of the trace metal iron in infectious diseases has been investigated in numerous clinical and animal studies. As in humans, iron is also essential for nearly all microorganisms, including bacteria, fungi, protozoa and helminths, with only a few exceptions [133]. In many bacteria, proliferation capacity and virulence is dictated by the amount of iron in their environment [134]. During infection, thus confined to the space of the host, bacteria must acquire iron from the host to sustain proliferation and accomplish efficient infection. Appropriately, systemic iron overload is associated with an increased risk of infection with various pathogens [135]. The human host has evolved several immune strategies to take advantage of this bacterial iron demand by sequestering iron from the specific localization in which a pathogen resides. The withdrawal of essential nutrients from pathogens, especially iron, is regarded as an efficient innate host defense strategy in line with the concept of innate nutritional immunity [6,136].

In systemic infection, the host adapts its iron metabolism to limit iron availability to pathogens, as well as attacking bacterial iron acquisition on multiple levels. Initiated by the production of pro-inflammatory cytokines during the acute-phase-response, several effector molecules are produced by the liver which affect systemic iron metabolism [135]. One prototypical mechanism leading to iron sequestration is the secretion of the hormone and master regulator of systemic iron metabolism hepcidin antimicrobial peptide (HAMP). HAMP binds to the only known ferrous iron exporter ferroportin-1 (FPN), leading to its internalization, degradation and thus, a decrease in iron export [137]. It is likely that most tissues are affected by this mechanism, with FPN degradation in hepatocytes, macrophages as well as duodenal enterocytes most likely contributing pivotally to hypoferremia and intracellular iron sequestration [138]. Transcriptional and post-transcriptional regulation of ferritin (as elaborated above) further increase intracellular iron storage capacity during systemic infection. This shift of iron into intracellular space is generally thought to increase host resistance against infection. Some bacteria though, accommodated to intracellular growth, may benefit from higher iron concentrations inside host cells [139].

Various studies provide evidence that in the light of specific (sub-) cellular localization of a pathogen, a differential response in terms of cellular iron metabolism is elicited in host cells [91,139,140,141]. This response depends on the primary intra- or extracellular localization of a pathogen and aims at starving the pathogen of iron, thus benefitting host defense [140,142]. In the case of infection with predominantly extracellular bacterial pathogens, the primary host strategy is to induce hypoferremia by upregulating cellular iron import and in parallel downregulating export [143]. In contrast, iron sequestration can be detrimental in the case of infection with intracellular pathogens, as the activation of the host response HAMP-FPN axis, which results in increased cellular iron sequestration, leads to enhanced intracellular pathogen growth [144]. Indeed, higher availability of iron in the pathogens’ cellular compartment is associated with increased growth in various models [145,146,147,148]. On this account, numerous studies have revealed differential cellular iron handling in the case of infection with intracellular pathogens. Specifically, the increase in iron export, mainly achieved by FPN induction, starves the pathogen and reduces intracellular bacterial proliferation [91,141,149,150,151]. This response is facilitated by at least two independent mechanisms: IFN-γ, primarily produced by activated natural killer cells and T-helper cells type 1, stimulates FPN expression and reduces TFR1 expression in infected macrophages [92,151]. During infection, nitric oxide species activate the transcription factor Nrf-2, which further promotes FPN induction [152].

Apart from FPN, NRAMP1 and DMT1 facilitate iron egress directly from the phagosome, making the metal less accessible for phagocytosed bacteria. Loss of these transporters alters cellular iron content and leads to higher iron availability to intracellular pathogens [153,154,155]. In the same line of reason, treatment with iron chelators has been shown to reduce intracellular bacterial growth, which may be therapeutically exploitable [144].

In addition to changes in systemic or cellular iron metabolism, the mammalian host produces effector molecules that directly compete with invading bacteria for iron, or inhibit bacterial iron uptake. Lactoferrin is one such example, binding iron with high affinity [156]. Exerting its effects mainly at mucosa, its bacteriostatic effects have also been evinced in an animal septicemia model [157]. Another compound at the center of iron-related host defense is the siderophore scavenger neutrophil gelatinase-associated lipocalin (lipocalin-2, NGAL). Produced by not only innate immune cells, but also epithelial cells, renal cells and hepatocytes, NGAL specifically attacks siderophore-dependent iron uptake, a major bacterial iron acquisition mechanism [158,159]. Siderophores (e.g., enterobactin) are secreted predominantly by gram-negative bacteria, bind iron in their environment with high affinity, and subsequently facilitate iron delivery to the microbe. Host-derived NGAL in turn binds and inactivates siderophores, thus disrupting bacterial iron acquisition and consequently stunting the pathogens’ growth [160,161]. Due to the key role of siderophores in a pathogen’s success, coevolution led to an immune evading mechanism of some bacterial species by producing alternative siderophores, which are not targeted by NGAL. Exemplary for this arms race, *Klebsiella pneumoniae* is capable of producing not only enterobactin, but also the alternative siderophores yersiniabactin and salmochelin. When this pathogen is challenged with host NGAL, expression of these alternative siderophores is an important virulence factor, enabling scavenging of iron despite the presence of NGAL and thus promoting infection [162].

Given the decisive impact of iron availability on bacterial infections, and the fundamental role ferritin provides in iron metabolism, its presence, regulation and distribution critically affect both the host’s and the pathogen’s success. In the main part of this review that follows, we will shine light on these factors and their implications for bacterial infection.

## 2. Main Part

Although the effect of iron on the course of infection has been investigated intensively [136], the fate of ferritin, as the main intracellular iron storage protein, during infection is less well understood.

### 2.1. Ferritin as a Bacterial Iron Source

Ferritin has been determined to be utilizable as an iron source by many different types of bacteria (Table 1), including *Yersinia pestis* [163], *Escherichia coli* [164,165], *Salmonella enterica* serovar Typhimurium [165], *Listeria monocytogenes* [166,167], *Burkholderia cenocepacia* [168], *Pseudomonas aeruginosa* [169], *Bacillus cereus* [170], *Streptococcus pyogenes* [171] *Vibrio vulnificus* [172], *Vibrio parahaemolyticus* [173] and *Mycobacterium tuberculosis* [174]. In these studies, chelation [164,165,169,170,174] as well as reduction of the iron core [165,166,169] have been proposed as the most common mechanisms for the acquisition of ferritin-bound iron by bacteria. Furthermore, bacterial proteases play a role in the mobilization of iron in *Pseudomonas aeruginosa* and *Burkholderia cenocepacia* [168,169]. Proteolytic degradation might also be important in other bacteria [175], but its effect on iron acquisition has either not been tested or is difficult to assess due to unexpected effects of protease inhibitors on bacterial iron metabolism [165]. Additionally, *Bacillus cereus* binds ferritin to its surface, which facilitates the acquisition of ferritin-bound iron [170]. If such a binding of ferritin is also present in other bacteria, has yet to be determined, but might also be relevant for bacteria like *Listeria monocytogenes*, which acquires iron from ferritin via surface-associated ferrireductases [166]. While ferritin is a sufficient iron source for many pathogens, it may still constitute a relevant obstacle for bacterial iron acquisition [165]. These studies demonstrate that bacterial pathogens from at least three phyla are able to utilize mammalian ferritin as sole iron source in vitro.

There are also some studies available, which assessed the interaction between ferritin and pathogens [176,177,178,179,180,181] (Table 1). Thereby, *Neisseria menigitidis* (*Nm*), *Mycobacterium bovis*, *Chlamydia trachomatis*, *Chlamydia pneumoniae* and *Helicobacter pylori* have been shown to colocalize with ferritin during intracellular infection [176,177,179,180]. Except for *Nm*, these findings are indicative of a direct extraction of ferritin-bound iron. Nevertheless, the exact mechanism of bacterial iron acquisition from ferritin has not been investigated in these studies. However, it is tempting to assume that mycobacteria may acquire ferritin-bound iron directly through such an association, as *Mycobacterium tuberculosis* has been shown to utilize ferritin as an iron source and to utilize extracellular as well as intracellular iron [174,182]. Nevertheless, there are studies which have investigated the role of ferritin as an iron source during infection with *Nm*, *Ehrlichia chaffeensis* (*Ech*), and uropathogenic Escherichia coli (UPEC) [178,180,181]. The study investigating UPEC showed that bacterial persistence in urothelial cells inside autophagosomes is facilitated by ferritinophagy and subsequent iron access to bacteria [178]. However, the authors also propose that UPEC is unable to sequester iron from ferritin. In contrast, a recent study demonstrated that another UPEC strain can use ferritin as a sole iron source in vitro [165]. This indicates that direct acquisition of ferritin-bound iron might also work for UPEC during infection. In the case of *Ech*, the pathogen secretes an effector protein, namely Ehrlichia translocated factor-3 (ETF-3), into the cytoplasm, which then binds to FTL and subsequently targets host iron stores to ferritinophagy [181]. Interestingly, *Nm* is not able to use ferritin as an iron source in vitro [183,184]. Still, *Nm* uses ferritin-derived iron as its main iron source during infection [180]. Thereby, it is speculated by the authors that *Nm* may trigger an iron starvation response and subsequently enhanced ferritinophagy, which enables *Nm* to acquire iron from degraded ferritin [180]. In conclusion, there is increasing evidence that ferritin may serve as an important bacterial iron source during infection, although hitherto only a small number of mechanisms have been identified. Unfortunately, most of the studies only focused on specific snippets of the whole picture, e.g., investigating whether a pathogen can use ferritin as its sole iron source or if a pathogen colocalizes with ferritin intracellularly, and were not followed up afterwards. Therefore, the role of ferritin at the host–pathogen interface remains largely unresolved for most of the investigated pathogens, emphasizing the need for further research.
ijms-24-04659-t001_Table 1Table 1List of investigated pathogens and the proposed mechanisms involved in pathogen–ferritin interaction. Question marks indicate that the mechanism has not been investigated in this species. However, it is to mention that specific mechanisms likely also work in other species which possess the same mechanism, e.g., same siderophore. a: assessed by fluorescence imaging; b: assessed by sucrose density gradient centrifugation.Bacterial Species InvestigatedPhylumFerritin Used as a Sole Iron SourceProposed Mechanisms for Pathogen-Ferritin InteractionSources*Mycobacterium tuberculosis*Actinomycetotayessiderophores[173]*Bacillus cereus*Bacillotayesbinding of ferritin via IlsA,siderophores[169]*Listeria monocytogenes*Bacillotayessurface-associated ferrireductases[165,166]*Streptococcus pneumonia*Bacillota?proteolysis[174]*Streptococcus pyogenes*Bacillotayes?[170]*Burkholderia cenocepacia*Pseudomonadotayesproteolysis[167]*Escherichia coli*Pseudomonadotayessiderophores[163,164]*Pseudomonas aeruginosa*Pseudomonadotayessiderophores,reduction of ferritin-bound iron,proteolysis[168]*Salmonella enterica* serovar TyphimuriumPseudomonadotayessiderophores,ferrous iron uptake systems[164]*Vibrio parahaemolyticus*Pseudomonadotayes?[172]*Vibrio vulnificus*Pseudomonadotayes?[171]*Yersinia pestis*Pseudomonadotayes?[162]**Investigations of****cell infection**



*Mycobacterium bovis*Actinomycetota?colocalization with ferritin ^a^[178]*Helicobacter pylori*Campylobacterota?colocalization with ferritin ^b^[175]*Chlamydia pneumoniae*Chlamydiota?colocalization with ferritin ^a^[176]*Chlamydia trachomatis*Chlamydiota?colocalization with ferritin ^a^[176]*Ehrlichia chaffeensis*Pseudomonadota?induction of ferritin via Etf-3[180]*Neisseria meningitidis*Pseudomonadotanoinduction of ferritinophagy via iron starvation of the host, colocalization with ferritin[179]Uropathogenic *Escherichia coli*Pseudomonadotayesiron acquisition via ferritinophagy[177]


### 2.2. Infection and Establishment of the Intraellular Replication Niche of Salmonella enterica serovar Typhimurium

*Salmonella enterica* serovar Typhimurium is a typical model microorganism for infection with an iron-dependent intracellular pathogen. Although ferritin has not been directly investigated as an in vivo iron source yet [185], the pathogen’s mechanisms of invasion, persistence and replication have been thoroughly investigated [186]. Furthermore, a recent study found that *S*.Tm is able to utilize ferritin-bound iron via its diverse iron uptake pathways [165]. Therefore, we herein use this infection model as a ground to discuss the implications on ferritin metabolism for infection with intracellular bacteria and try to draw parallels to the diverse investigations with other model pathogens. In particular, we will focus on the Salmonella pathogenicity island (SPI)-1 mediated infection, because it is the most commonly investigated mode of infection. However, it is necessary to mention that there is a difference in the expression of virulence genes encoded on SPI-1 and SPI-2, depending on whether *S*.Tm enters the cell actively via SPI-1 or becomes phagocytosed, which can be facilitated by opsonization [187]. An extensive review on the intracellular processes during infection with *S*.Tm is provided by Knuff et al. However, we will go through the relevant steps below [186].

SPI-1 and SPI-2 encode syringe-like type 3 secretion system (T3SS), which *S*.Tm uses to transport effector proteins across membranes of host cells [186]. Prior to infection, *S*.Tm injects a cocktail of effector proteins via the SPI-1 encoded T3SS into the cytoplasm, which results in membrane ruffling and subsequent invasion of the pathogen into an intracellular membranous compartment termed the *Salmonella*-containing vacuole (SCV) [188,189]. Just after invasion, the early SCV is characterized by markers of the early endosome such as early endosome antigen-1 (EEA1) and Ras-related proteins Rab4, Rab5, Rab11 as well as TFR1 [190,191,192]. When *S*.Tm enters the cell using SPI-1, the SCV is damaged by the T3SS [193]. This damage results in recruitment of galectin-8 to the SCV [194]. The damaged SCV can then be repaired by *S*.Tm by utilizing the host autophagy machinery [195]. This repair is necessary for the activation of SPI-2 [195]. Upon maturation to the intermediate SCV, the vacuole loses the markers of the early endosome and acquires markers of the late endosome such as Lysosomal-associated membrane protein (Lamp)-1, Lamp-2, Lysosome integral membrane protein 1 (Limp-1, CD63), vATPase and Rab7 [190,196,197]. The effector proteins of SPI-2 then induce the final maturation step to the late SCV, which is associated with an extensive tubular system including the Salmonella-induced filaments (SIF) [198,199]. The formation of the SIF represents the completed establishment of the intracellular replication niche of *S*.Tm and coincides with enhanced metabolic activity and bacterial replication [198,199].

### 2.3. Implications on Ferritin Metabolism upon Infection with Salmonella Typhimurium

During infection with *S*.Tm, several host or pathogen-driven mechanisms may affect ferritin metabolism. These are depicted in Figure 3. Starting just after invasion, the SCV associates with the known ferritin receptor TFR1, which indicates that *S*.Tm may have access to ferritin from the onset of infection [119]. However, the damaged early SCV also recruits galectin-8, which may trigger secretory autophagy, in an effort of the host cell to prevent utilization of ferritin via a compartment shift [117]. Furthermore, this mechanism might be a more general one, because T3SS, which is responsible for the membrane damage, as well as other syringe-like secretion systems are employed by a large variety of bacterial pathogens [200]. Additionally, failure to repair such damage or to sustain the SCV results in cytosolic hyper-replication in permissive cells, e.g., epithelial cells. This might be facilitated by direct contact with cytoplasmic ferritin, which has been shown to significantly enhance bacterial growth of *S*.Tm [165,195,201]. In this context, cytoplasmic ferritin might also be a rich iron source for pathogens escaping their vacuolar compartment, such as *Listeria monocytogenes*, which has been shown to release ferritin-bound iron via surface-associated reductases in vitro [166,202]. The SCV enables *S*.Tm to acquire endosomal cargo from endocytosis-derived as well as intracellular vesicles [199]. Whether this is specific for *S*.Tm or might also be the case for other intracellular pathogens, which also reside in an endosomal/lysosomal-like compartment, has yet to be determined [203]. Given that some studies of such pathogens found a colocalization of bacteria with ferritin, this might be a more general mechanism in intracellular infections [176,177,179,180]. The uptake of endosomal cargo is enhanced by SPI-2 and even further with SIF, and is thought to provide nutrients for the replication of *S*.Tm [199]. Furthermore, this finding indicates that intracellular ferritin, which is trafficked via vesicles to the lysosome, might be delivered directly to *S*.Tm. Additionally, extracellular ferritin, which enters the cell via receptor-mediated endocytosis, might also be directed to the SCV. Moreover, the delivery of ferritin to the SCV might be enhanced depending on the efficiency of the uptake of extracellular iron via this mechanism, as the prevention of iron uptake from extracellular sources would result in induction of ferritinophagy by reducing the labile iron pool (LIP) in a similar manner as is proposed during infection with *Nm* [180]. A possible strategy of the host to counter this mechanism might be the utilization of DMT1. DMT1 has been shown to be expressed on the plasma membrane of macrophages and to transport iron directly form the extracellular space to the cytoplasm without the need for it to pass through an endosomal compartment [204]. This can be facilitated by local hypoxia, which increases DMT1 expression and local tissue acidosis, which improves the iron transport capacity of DMT1 [205,206,207,208]. Thereby, the LIP might be maintained just high enough to prevent ferritinophagy, but low enough to avoid enhanced bacterial growth via a concomitant upregulation of FPN [209]. The upregulation of FPN might then also trigger FPN-dependent mobilization of ferritin-bound iron and target ferritin for proteasomal degradation [104], which might represent a host adaptation against intracellular vacuolar pathogens. Furthermore, this might be an additional reason why nifedipine, which has been shown to increase DMT1-dependent iron uptake and subsequently promotes FPN expression and cellular iron egress, acts beneficially for the host in the case of infection with *S*.Tm [207,209]. Noteworthily, *S*.Tm aims to reduce *Fpn1* transcription via its SPI-2 effector protein SpvB, which induces the proteasomal degradation of Nrf-2 [210]. Another possibility by which the host might utilize a compartment shift of ferritin is by secretion of ferritin via CD63-positive exosomes [115]. In line, a recent study found that macrophages secrete CD63-positive exosomes upon *S*.Tm-induced endoplasmic reticulum (ER) stress and subsequent lysosomal dysfunction [211]. Furthermore, these extracellular vesicles (EV) are enriched with TFR1, CD91 and CD163 to scavenge iron from the circulation [211]. However, it is hitherto not known if M1-derived EVs contain ferritin. Nevertheless, the fact that ferritin secretion via exosomes is CD63-dependent and CD63 expression is also regulated via the IRE/IRP system indicates that this might be the case [115,212].

In summary, there are many possibilities of how *S*.Tm might modulate ferritin metabolism to its advantage, but also numerous ways by which the host might counteract these efforts. Thereby, some of the mechanisms may be of a more general, intrinsic nature of intracellular bacterial infections, while others are *Salmonella* specific.

### 2.4. Outlook on Future Research

The recent advances in our understanding of ferritin metabolism and the ever-expanding knowledge surrounding the pathomechanisms of bacterial infection open up many possible ways to investigate the role of ferritin at the host–pathogen interface. Although the proposed mechanisms, based on an infection with *S*.Tm, are hitherto in the realm of speculation, they are exemplarily for the branches future research may follow. Thereby, an important part of the whole picture would be to investigate the different modes of release of ferritin-bound iron. As explained above, cytoplasmic iron release and subsequent degradation via the proteasome, hitherto scarcely investigated, might be an important mechanism during infection. In this context, a knockout of TAX1BP1 could bring valuable insights, because it would supposedly abolish vesicular ferritin-trafficking and, as a consequence, ferritinophagy, while preserving the subcellular location of ferritin inside a liquid-phase condensate [93]. However, such a knockout would result in a different problem, as TAX1BP1 is also involved in the clearance of pathogens via xenophagy [213]. Another important aspect to investigate would be the role of CD63, as it is also regulated via the IRE/IRP system and is majorly involved in secretion via the MVB-exosome system. Thus, it might bring valuable insights into the route vesicular ferritin takes, be it secretion or degradation. A recent study found an increase of CD63-positive exosomes in serum during infection with *S*.Tm and *Staphylococcus aureus* in mice [211]. Furthermore, these exosomes possess TFR1, CD163 and CD91 on their surface and sequester iron from serum [211]. Moreover, this secretion is induced by ER stress and subsequent lysosomal dysfunction, which may be an important signal for secretion rather than degradation [211]. This is of interest because ER stress is a common feature found during many bacterial infections and has also been found in lipopolysaccharide (LPS)-induced inflammation, and thereby it might represent a more general host defense mechanism [214,215]. However, it has not been investigated if those EV also contain ferritin. Nevertheless, other proteins which may impede iron acquisition from ferritin, such as NGAL and superoxide dismutase, have also been found in extracellular vesicles [165,216,217,218]. Hence, it would be interesting to see if such proteins are also found in ferritin-containing exosomes. Further research should also investigate the role of galectin-8 and endo/lysosomal-like bacteria-containing compartments, because they represent mechanisms common during many intracellular infections [194,203]. Although the vesicular compartments differ from pathogen to pathogen, many associate with markers of the endo/lysosomal system, which might indicate that such vacuolar compartments have the intrinsic ability to acquire endosomal cargo via vesicle fusion, which is supported by the colocalization of ferritin with a variety of different pathogens [176,177,179,180].

## 3. Conclusions

Recent years have brought much insight into the mechanistic details of ferritin regulation, metabolism, secretion and uptake. This enables a more systematic approach for further research, whereby it will be easier to investigate and target specific mechanisms of ferritin metabolism in different situations. However, there is still much to learn, as specific branches of ferritin metabolism have not been followed up very much. For example, the diverse functions of ferritin at the crossroads of intracellular iron sequestration and immune metabolism in the context of infection have mostly been investigated secondarily. Nevertheless, a few studies have dared to directly investigate ferritin at the host–pathogen interface and could soundly demonstrate that ferritin is an important iron source for some bacteria. We are confident that advances in the knowledge of ferritin metabolism will facilitate future research and as a consequence provide a more holistic picture of nutritional immunity and the role of ferritin in infection and inflammation.

## Figures and Tables

**Figure 1 ijms-24-04659-f001:**
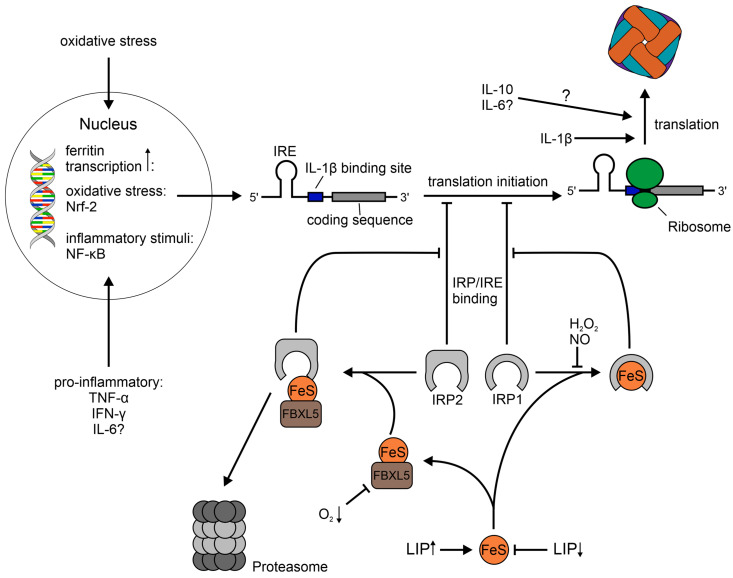
Simplified schematic depiction of regulatory mechanisms influencing ferritin expression. Ferritin transcription has been shown to be induced by oxidative stress and the pro-inflammatory cytokines TNF-α and IFN-γ via their downstream signals. IL-6 is associated with increased ferritin protein expression. However, it is not established whether this happens on a transcriptional or post-transcriptional level. The ferritin mRNA transcript possesses regions in it 5′ UTR, namely IRE, an IL-1β-binding site and a binding site for eIF3 (not shown), which affect the translation of ferritin. If the cellular iron content is low, the iron–sulfur cluster synthesis is reduced, which enables IRP1 and IRP2 to bind the IRE on ferritin mRNA and subsequently block the translation. If the cellular iron content is high, IRP1 moves from the IRE, while binding an iron-sulfur cluster, becoming cytoplasmic aconitase. Furthermore, FBXL5 binds an oxygen-responsive iron–sulfur cluster and mediates the proteasomal degradation of IRP2. The IRP1/IRE binding is enhanced by H_2_O_2_ and NO, while the effects of other ROS and RNS on the IRE-binding capability of IRP1 and IRP2 are hitherto unclear (not shown). IL-10 increases ferritin protein expression on a post-transcriptional level. However, the specific mechanism is hitherto not resolved. The translation of ferritin is enhanced in the presence of IL-1β by binding to a guanine and cytosine-rich domain in the 5′-UTR of ferritin mRNA. Hitherto unknown or unresolved mechanisms are highlighted with a question mark. Nrf-2: nuclear factor erythroid 2-related factor 2, NF-κB: nuclear factor κB TNF-α: tumor necrosis factor α, IFN-γ: interferon γ, IL: interleukin, RNS: reactive nitrogen species, ROS: reactive oxygen species, IRP: iron regulatory protein, IRE: iron-responsive element, LIP: labile iron pool, FBXL5: F-box and leucine-rich repeats protein 5, FeS: iron–sulfur cluster, eIF3: eukaryotic translation initiation factor 3.

**Figure 2 ijms-24-04659-f002:**
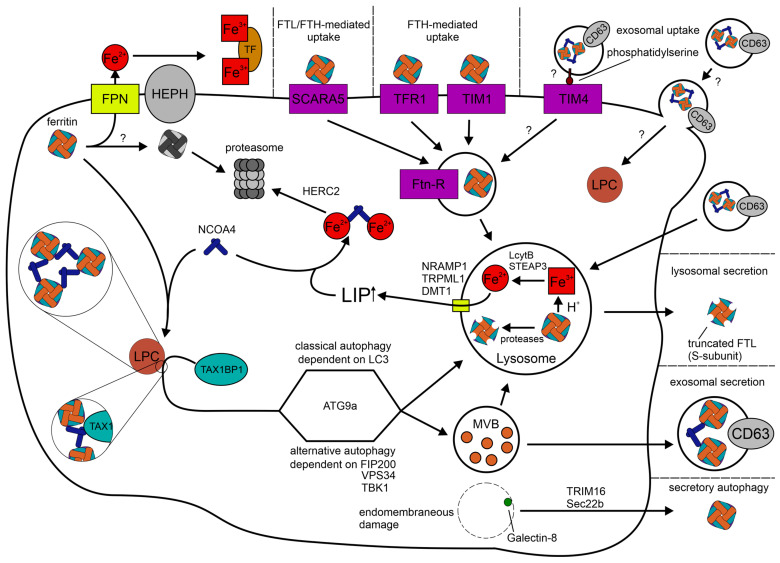
Schematic depiction of ferritin metabolism and trafficking in a hypothetical human cell. Known and putative receptors involved in ferritin uptake are marked in violet. SCARA5 is also representative for SCARA1 and SCARA2, which have a lower affinity for ferritin. Ferrous iron transporters are marked in yellow. Ferritin uptake mechanisms are depicted on the top right and mechanisms for ferritin secretion are shown on the right. Inside the cell, the process of ferritinophagy, which is the trafficking of ferritin to the lysosome and subsequent iron release from ferritin, as well as the FPN mediated proteasomal degradation are depicted. Hitherto unknown or unresolved mechanisms are highlighted with a question mark. FPN: ferroportin-1, NCOA4: nuclear receptor coactivator 4, LPC: liquid-phase condensate, TAX1BP1: Tax1-binding protein 1, FTL: ferritin L-subunit, FTH: ferritin H-subunit, SCARA5: Scavenger receptor class A member 5, TFR1: transferrin receptor 1, TIM: T-cell immunoglobulin and mucin domain, Ftn-R: ferritin receptor, HERC2: HECT domain and RCC1-like domain-containing protein 2, LC3: Microtubule-associated protein 1A/1B-light chain 3, FIP200: FAK family kinase-interacting protein of 200 kDa, VPS34: Phosphatidylinositol 3-kinase VPS34, TBK1: TANK-binding kinase 1, TRIM16: Tripartite motif-containing protein 16, Sec22b: Vesicle-trafficking protein SEC22b, MVB: multivesicular body, NRAMP1: Natural resistance-associated macrophage protein 1, STEAP3 six-transmembrane epithelial antigen of prostate 3, and LyctB: lysosomal cytochrome B, DMT1: Divalent metal transporter 1, TRPML1: Transient receptor potential channel mucolipin 1, LIP: labile iron pool.

**Figure 3 ijms-24-04659-f003:**
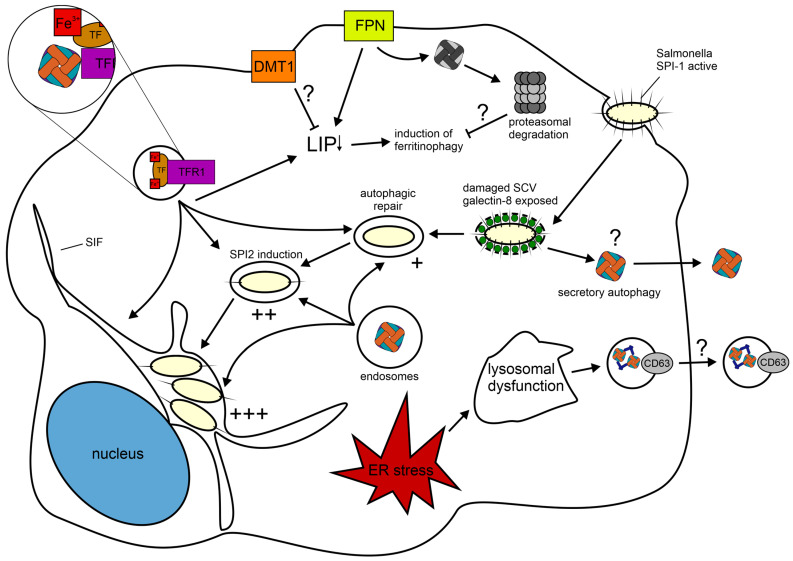
Schematic depiction of possible *S*.Tm or host-derived mechanisms regarding ferritin during infection. The graphic illustrates the path of the establishment of the intracellular replication niche of *S*.Tm. For simplicity, the vacuolar escape of *S*.Tm is not depicted, but it is necessary to mention that cytosolic *S*.Tm might use iron from ferritin-NCOA4 liquid-phase condensates. However, the entry of *S*.Tm via its SPI-1 encoded T3SS results in membrane damage of the SCV, which might trigger secretory autophagy of ferritin. Subsequently, *S*.Tm repairs this damage via the host autophagy machinery. Inside the SCV, *S*.Tm is able to acquire endosomal cargo, which likely includes ferritin. The access to endosomal cargo is enhanced by SPI-2 and even further upon the formation of SIF. The ability of the differently maturated SCV to acquire endosomal cargo is marked with increasing plus signs adjacent to the respective SCV. Furthermore, this endosome hijacking by *S*.Tm might result in reduced intracellular iron levels by inhibiting uptake of extracellular iron, which would consequently trigger ferritinophagy to satisfy the host’s iron demand. A possible host defense strategy against this might be direct uptake of iron via DMT1. Additionally, FPN might mobilize ferritin-bound iron, which may result in the ensuing proteasomal degradation of iron-poor ferritin in the cytosol. Another host defense strategy might also be a shift of ferritin to the extracellular compartment via the secretion of ferritin-containing exosomes. Thereby, *S*.Tm induced ER stress, which in turn results in lysosomal dysfunction and the subsequent release of CD63-positive exosomes, which might also contain ferritin. The SPI-1 and SPI-2 encoded T3SS are depicted as spikes on *S*.Tm. Hitherto unknown or unresolved mechanisms are highlighted with a question mark. SPI-1/2: Salmonella pathogenicity island, T3SS: type 3 secretion system, SCV: Salmonella-containing vacuole, TFR1: Transferrin receptor 1, TF: Transferrin, ER: endoplasmic reticulum, FPN: ferroportin-1, SIF: Salmonella-induced filaments, NCOA4: nuclear receptor coactivator 4.

## Data Availability

Not applicable.

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
