# Peer review of "Advances in Ferritin Physiology and Possible Implications in Bacterial Infection"

_ijms, 2023, doi:10.3390/ijms24054659_

Round 1

Reviewer 1 Report

Tittle: Advances in ferritin physiology and possible implications for bacterial infection

The manuscript by Gehrer et al. discusses recent updates on ferritin’s functions and its implications on host-pathogen, especially in bacterial infection. Overall, the manuscript is well written and requires revision before its publication as follows:

Comments

1.     Lines 30-32, Some more details on various defense systems can be provided (minor), and the status of bacterial infection issues/mortality. During bacterial infection, the host adapts different strategies or vice versa, such as quenching of signal molecules that requires by bacterial pathogens for its multiplication/release of toxins (quorum sensing) i.e., Lancet 400 (2022) 2221–2248, Biotechnology advances 37 (2019) 62-90; Molecules 27 (2022) 7584.

2.     The authors mainly stated the qualitative information; the detailed quantitative discussion can be provided in most of the sections.

3.     Please add at least one brief Table on bacterial infections and the role of ferritin based on literature.  

Reviewer 2 Report

The review provides good information about ferritin physiology. However, I found that the title is misleading and the review itself doesn't give any clear message regarding the use of ferritin metabolism against bacterial infections. Although the authors use 'possible implications', the review will fail to convince potential readers of the connection between ferritin and bacterial infections.

In addition, the authors should give a figure of the structure of the ferritin and indicate all its structural features.

There is no mention of the role of mini-ferritins in bacteria.  Mini-ferritins are used to protect bacteria from iron. How would mini-ferritins interfere in these settings? 

Round 2

Reviewer 1 Report

Accept as is.

Reviewer 2 Report

The authors have tried to improve the text and make their hypotheses more clear. The new text reads better. I hope potential readers would find it useful.